# Impacts of Soil Properties on Species Diversity and Structure in *Alternanthera philoxeroides*-Invaded and Native Plant Communities

**DOI:** 10.3390/plants13091196

**Published:** 2024-04-25

**Authors:** Hao Wu, Yuxin Liu, Tiantian Zhang, Mingxia Xu, Benqiang Rao

**Affiliations:** 1College of Life Sciences, Xinyang Normal University, Xinyang 464000, China; liuyuxin141022@163.com (Y.L.); zhangtian1103@163.com (T.Z.); 15839642920@163.com (M.X.); 2Henan Dabieshan National Field Observation and Research Station of Forest Ecosystem, Zhengzhou 450046, China; 3Xinyang Academy of Ecological Research, Xinyang 464000, China; 4Dabie Mountain Laboratory, Xinyang 464000, China

**Keywords:** habitat heterogeneity, biological invasions, plant community, diversity index, *Alternanthera philoxeroides*, species coexistence

## Abstract

Soil properties can affect plant population dynamics and the coexistence of native and invasive plants, thus potentially affecting community structure and invasion trends. However, the different impacts of soil physicochemical properties on species diversity and structure in native and invaded plant communities remain unclear. In this study, we established a total of 30 *Alternanthera philoxeroides*-invaded plots and 30 control plots in an area at the geographical boundary between North and South China. We compared the differences in species composition between the invaded and native plant communities, and we then used the methods of regression analysis, redundancy analysis (RDA), and canonical correspondence analysis (CCA) to examine the impacts of soil physicochemical properties on four α-diversity indices and the species distribution of these two types of communities. We found that *A*. *philoxeroides* invasion increased the difference between the importance values of dominant plant species, and the invasion coverage had a negative relationship with the soil-available potassium (*R*^2^ = 0.135; *p* = 0.046) and Patrick richness index (*R*^2^ = 0.322; *p* < 0.001). In the native communities, the species diversity was determined with soil chemical properties, the Patrick richness index, the Simpson dominance index, and the Shannon–Wiener diversity index, which all decreased with the increase in soil pH value, available potassium, organic matter, and ammonium nitrogen. However, in the invaded communities, the species diversity was determined by soil physical properties; the Pielou evenness index increased with increasing non-capillary porosity but decreased with increasing capillary porosity. The determinants of species distribution in the native communities were soil porosity and nitrate nitrogen, while the determinants in the invaded communities were soil bulk density and available potassium. In addition, compared with the native communities, the clustering degree of species distribution in the invaded communities intensified. Our study indicates that species diversity and distribution have significant heterogeneous responses to soil physicochemical properties between *A*. *philoxeroides*-invaded and native plant communities. Thus, we need to intensify the monitoring of soil properties in invaded habitats and conduct biotic replacement strategies based on the heterogeneous responses of native and invaded communities to effectively prevent the biotic homogenization that is caused by plant invasions under environmental changes.

## 1. Introduction

Plant invasions seriously threaten the function and biodiversity of the global ecosystem [1] and cause the loss of native functional groups [2]. Alien plants can successfully invade due to their strong phenotypic plasticity and interspecific competitiveness and through interactions with soil, microorganisms, and native plants [3,4]. For example, the widespread invasion of *Solidago canadensis*, *Solanum elaeagnifolium*, and *Ageratina adenophora* has significantly decreased the species diversity of native plant communities [5,6]. The invasive species *Phragmites australis* in Australia weakens the competitiveness of native plants, and plant diversity decreases with increasing *P*. *australis* density [7]. The invasive species *Imperata cylindrica* has a higher water use efficiency than the accompanying species, thus inhibiting the latter’s growth and reproduction [2]. Increased *Lupinus polyphyllus* invasion coverage intensifies the biohomogenization of native plant communities [8]. *Impatiens glandulifera* can enhance its invasive competitiveness by reducing the number of beneficial epiphytic fungi on native accompanying plants [9]. Due to its great niche breadth, long flowering period, large seed quantity, and high root–shoot ratio, *Gaillardia aristata* can transform from an alien ornamental species into an invader under stressful environmental conditions, and it decreases the plant richness and diversity indices in invaded habitats [10]. Thus, exploring the correlation between plant invasions and native species diversity, as well as their environmental determinants, is beneficial for predicting community succession dynamics and elucidating species coexistence mechanisms in invaded ecosystems.

Soil is the main source of plant nutrition, and its physicochemical properties and resource availability affect plant physiological and ecological characteristics [11], which further affect plant population dynamics and species diversity [12,13]. The spread of invasive plants and their effects on native communities are also regulated by soil properties [14,15,16]. For example, high organic matter content in soil can increase the population density and root numbers of the invasive species *Ludwigia grandiflora* and thus exacerbate its invasion process [17]. High soil nitrogen can increase the abundance, height, biomass, and seed quantity of the invasive species *Mikania micrantha* and *Bidens pilosa*, thereby weakening native plant diversity [18,19]. Heterogeneous soil nutrients can promote the aboveground biomass accumulation of the invasive species *Ageratum conyzoides*, *B*. *pilosa*, and *Erigeron annuus* [13]. Plant invasions can also alter the soil conditions in microhabitats. For example, *Chromolaena odorata* invasion causes a significant increase in soil nitrogen, available phosphorus, and pH value, which promote the biomass accumulation of subsequent invasive plants, although this promoting effect is weakened by the increase in native plant richness [16]. *Agerina adenophora* invasion leads to an increase in soil nitrogen and a decrease in pH, which inhibit the seed germination and seedling growth of native plants [20]. However, studies that examine the impacts of soil physicochemical properties on invaded and native plant communities at the same spatiotemporal scale are relatively rare.

*Alternanthera philoxeroides*, also known as alligator weed, a noxious invasive weed native to South America belonging to the Amaranthaceae family, grows in both aquatic and terrestrial environments [21]. *A*. *philoxeroides* has a robust asexual reproductive ability, which enables it to propagate asexually through the fracture of stem segments and adventitious roots on stem nodes, and thus, it rapidly spreads and establishes populations across diverse habitats [21]. Furthermore, *A*. *philoxeroides* can produce a variety of secondary metabolites, including alkaloids, organic acids, and flavonoids, to inhibit the growth of other accompanying plants, providing favorable conditions for its diffusion and establishment [22]. Due to its low genetic diversity, strong phenotypic plasticity, and fast growth rate, *A*. *philoxeroides* has invaded various parts of the world, including the United States, China, South Africa, India, and Australia, causing damage to the ecological environment, biodiversity, and economic development [22,23]. In China, *A*. *philoxeroides* has widely invaded more than 20 provinces and is included in the list of 59 invasive alien species under key management, and climate change threatens to further exacerbate the damage of *A*. *philoxeroides* to native plants at higher latitudinal regions [24]. Previous studies have found that *A*. *philoxeroides* can enhance its photosynthesis rate by increasing its nitrogen absorption rate and resource utilization, seriously inhibiting native plant growth in interspecific competition [12]. In environments with high heterogeneity, *A*. *philoxeroides* has a stronger clonal integration ability than native plants [25], and the high phenotypic plasticity increases this invader’s tolerance to heavy metal stress, thus enhancing its competitiveness against native plants [26]. *A*. *philoxeroides* can reduce soil enzyme activity and disrupt the balance of soil microbial communities through allelopathic effects, thus decreasing the abundance of microorganisms that are beneficial to native plants [27]. The performance of *A*. *philoxeroides* is also influenced by soil factors; e.g., soil nitrogen addition can increase the stem length and biomass accumulation of *A*. *philoxeroides*, and its invasion trend is positively correlated with nitrogen concentration [28,29]. The most abundant soil nutrients, such as phosphorus and potassium, can promote the secretion of more allelopathic substances in *A*. *philoxeroides* roots, causing more intense interspecies competition [27,30]. The height, biomass, and chlorophyll of *A*. *philoxeroides* all decrease with decreased soil pH [31]. Under high nutrient levels, an increase in the soil N:P ratio can significantly inhibit *A*. *philoxeroides* growth [32]. In addition, *A*. *philoxeroides* invasion causes an increase in soil total carbon, nitrogen, and water content and a decrease in soil N:P and C:P ratios [28,33,34].

However, the impacts of soil properties on species diversity and distribution in *A*. *philoxeroides*-invaded and non-invaded plant communities are currently unclear, although this is crucial for elucidating the ecological mechanisms of *A*. *philoxeroides* invasions during environmental changes. In this study, we investigated herb communities that are invaded by *A*. *philoxeroides* (Figure 1) and the native communities without *A*. *philoxeroides* at the geographical boundary between North and South China. We hypothesize that there are differences in the ecological effects of soil physicochemical properties on plant diversity and species distribution between these two types of communities. Specifically, we addressed the following questions: (1) What is the coupling relationship between species diversity and soil properties in invaded and native plant communities? (2) What are the limited soil factors that determine species distribution in these two types of communities?

## 2. Results

### 2.1. Plant Species Composition and Invasion Trend

In total, 114 plant species belonging to 49 families and 103 genera were recorded in 30 control plots; the families with the highest richness were Asteraceae (16 genera; 17 spp.), Poaceae (13 genera; 13 spp.), and Leguminosae (6 genera; 8 spp.). A total of 108 plant species belonging to 42 families and 96 genera were recorded in 30 invaded plots; the families with the highest richness were Poaceae (17 genera; 17 spp.), Asteraceae (15 genera; 17 spp.), and Leguminosae (8 genera; 8 spp.). The dominant species (total IV > 0.5) in the control plots were *Oxalis articulata*, *Cynodon dactylon*, *Eleusine indica*, *Galium odoratum*, *Duchesnea indica*, *Dichondra micrantha*, etc.; *O*. *articulata*, *C*. *dactylon*, and *E*. *indica* were the constructive species in the control communities (Table 1). The dominant species (total IV > 0.5) in invaded plots were *A*. *philoxeroides*, *Cynodon dactylon*, *Galium odoratum*, *Oxalis articulata*, *Eleusine indica,* etc., among which, the total IV of *A*. *philoxeroides* was much higher than that of other species, showing the dominance of this invader as the monodominant constructive species (Table 1). Compared with the control plots, *A*. *philoxeroides* invasion increased the difference between the total IV of dominant species in the invaded plots, and only a few species that had strong ecological adaptability, such as *C*. *dactylon*, *G*. *odoratum*, *O*. *articulata*, and *E*. *indica*, could continue maintaining the highest dominance in invaded communities (Table 1). Regression analysis showed that *A*. *philoxeroides* invasion coverage significantly decreased with an increase in soil AK (*R*^2^ = 0.135 and *p* = 0.046; Figure 2A) and the Patrick richness index (*R*^2^ = 0.322 and *p* < 0.001; Figure 2B).

### 2.2. RDA Ordination of Soil Properties and Species Diversity

In the control plots, the cumulative percentage of variance explaining the soil properties–species diversity relationships of the first two RDA axes reached 99.9% (94.5% for axis 1 and 5.4% for axis 2), and the first axis had the decisive effect (Figure 3A). The dominant soil factors that determined RDA axis 1 were soil pH value (*r* = −0.433; *p* < 0.05), AK (*r* = −0.418; *p* < 0.05), ORG (*r* = −0.400; *p* < 0.05), and N-NH_4_ (*r* = −0.379; *p* < 0.05), and the values of these four soil factors gradually decreased from left to right along RDA axis 1 (Table 2; Figure 3A). The four α-species diversity indices all showed a strong negative relationship with soil pH, AK, ORG, and N-NH_4_ (Figure 3A). The determinant of RDA axis 2 was soil AP (*r* = 0.634; *p* < 0.01), and it had a strong negative relationship with the Simpson dominance index, the Pielou evenness index, and the Shannon–Wiener diversity index (Table 2; Figure 3A).

In the invaded plots, the accumulated percentage explanation of the first two RDA axes reached 100.0% (99.8% for axis 1 and 0.2% for axis 2); the first axis determined the relationship between soil properties and species diversity (Figure 3B). There was no significant correlation between any of the soil factors and RDA axis 1 (Table 2). The determinants of RDA axis 2 were soil pH value (*r* = 0.677; *p* < 0.01), NCP (*r* = 0.629; *p* < 0.01), and CP (*r* = −0.594; *p* < 0.01) (Table 2). Among these, the Pielou evenness index had a strong positive relationship with soil pH and NCP and a strong negative relationship with CP (Figure 3B).

### 2.3. Regression Analysis of Soil Properties and Species Diversity

In the control plots, the optimal fitting relationship between CWC and the Patrick richness index was a cubic equation (*R*^2^ = 0.262; *p* = 0.017); with increasing CWC, the Patrick richness index first increased but then decreased (Figure 4A). The variation in the Patrick richness index (*R*^2^ = 0.140; *p* = 0.041), the Simpson dominance index (*R*^2^ = 0.147; *p* = 0.036), and the Shannon–Wiener diversity index (*R*^2^ = 0.159; *p* = 0.029) along the soil N-NH_4_ gradient was characterized by inverse equations; these three diversity indices all decreased with increasing N-NH_4_ (Figure 4B–D). The optimal fitting relationships between soil AP and Simpson dominance index are represented by a compound equation (*R*^2^ = 0.136; *p* = 0.045); increasing soil AP decreased the Simpson index (Figure 4E). These were all consistent with the results of RDA ordination (Figure 3A).

In the invaded plots, NCP presented a significant logarithmic regression relationship with the Pielou evenness index (*R*^2^ = 0.158; *p* = 0.029); the Pielou index increased with an increase in NCP (Figure 4F). Increasing CP significantly decreased the Pielou evenness index, which is shown by a linear equation (*R*^2^ = 0.152; *p* = 0.033) (Figure 4G). These were all consistent with the results of RDA ordination (Figure 3B). However, in the control plots, the impact of CWC on species diversity was not consistent between RDA ordination and regression analysis, and this inconsistent impact also applies to the soil pH in the invaded plots.

### 2.4. CCA Ordination of Soil Properties and Species Distribution

In the control plots, the cumulative percentage of variance explaining the soil properties–species distribution relationships of the first two CCA axes was 39.5% (23.2% for axis 1 and 16.3% for axis 2) (Figure 5A). The dominant soil factors that determined axis 1 were CP (*r* = 0.601; *p* < 0.01), N-NO_3_ (*r* = −0.463; *p* < 0.01), and ORG (*r* = −0.406’ *p* < 0.05); from left to right along axis 1, CP gradually increased, while soil N-NO_3_ and ORG gradually decreased (Table 3, Figure 5A). The determinants of axis 2 were NCP (*r* = 0.835; *p* < 0.01), TP (*r* = 0.791; *p* < 0.01), FWC (*r* = 0.772; *p* < 0.01), and ORG (*r* = 0.701; *p* < 0.01). These four soil factors all gradually increased from bottom to top along axis 2 (Table 3; Figure 5A). *O*. *articulata* (1) tended to be distributed in areas with high NCP and ORG; *C*. *dactylon* (2), *Digitaria sanguinalis* (9), and *Imperata cylindrica* (15) were distributed in areas with high CP but low N-NO_3_; *E*. *indica* (3) was distributed in areas with low ORG and N-NO_3_; *Duchesnea indica* (5) was distributed in areas with high N-NO3; and many species, including *G*. *odoratum* (4), tended to be distributed in areas with low FWC and TP (Figure 5A).

In the invaded plots, the first two CCA axes had an accumulated percentage explanation of 38.9% (22.8% for axis 1 and 16.1% for axis 2) (Figure 5B). The soil factors that determined axis 1 were soil AK (*r* = 0.435; *p* < 0.05), BD (*r* = −0.419; *p* < 0.05), and N-NH_4_ (*r* = −0.398; *p* < 0.05); from left to right along axis 1, AK gradually increased, while BD and N-NH_4_ gradually decreased (Table 3; Figure 5B). The determinants of axis 2 were CP (*r* = 0.737; *p* < 0.01), CWC (*r* = 0.706; *p* < 0.01), TP (*r* = 0.614; *p* < 0.01), and FWC (*r* = 0.525; *p* < 0.01). These four soil factors gradually increased from bottom to top along axis 2 (Table 3 and Figure 5B). *Beckmannia syzigachne* (20) tended to be distributed in areas with high BD but low AK and FWC; *Cyperus rotundus* (15) was distributed in areas with high N-NH_4_; *Artemisia argyi* (11) and *Dichondra micrantha* (17) were distributed in areas with low CWC, CP, and TP; and *A*. *philoxeroides* (1), *O*. *articulata* (4), *E*. *indica* (5), and *Rumex acetosa* (6) were all located at the center of the CCA ordination diagram, indicating that they all had wide ecological adaptability and superior growth advantages based on the same resource combination (Figure 5B). *Paspalum paspaloides* (8) and *Hydrocotyle sibthorpioides* (12) were far from the soil vectors, indicating that their distribution was weakly influenced by soil factors (Figure 5B). Compared with the control plots, the clustering degree of species distribution in the invaded plots increased (Figure 5).

## 3. Discussion

### 3.1. Mechanism of Impact of Soil on Community Structure and Plant Invasions

In natural communities, each species has the inherent ecological niche to achieve coexistence [35]. In our study, we found that *O*. *articulata* had the highest IV in control communities because, as an alien noninvasive species, it has strong stressful resistance and a strong allelopathic effect that inhibits the growth of native plants, which is beneficial to expanding the living space and achieving rapid growth [36,37]. The IV of *A*. *philoxeroides* was much higher than that of other accompanying species in the invaded communities, possibly due to its strong clonal reproductive ability and phenotypic plasticity, allowing this invader to plunder resources on a large scale, enhance competitiveness, and thus suppress native plants [22,24,38]. The IVs of some dominant species, such as *O*. *articulata*, *C*. *dactylon*, *E*. *indica*, and *G*. *odoratum*, decreased in the invaded communities compared with those in the control communities. Moreover, several native species with weaker tolerance, such as *Axonopus compressus*, and *Clinopodium chinense*, were squeezed out of the dominant layer, which indicates that *A*. *philoxeroides* invasion intensified interspecific competitions and accelerated the process of community homogenization [39].

The biotic resistance hypothesis proposes that native communities with high species richness are less susceptible to plant invasions because available niches are already filled, and invasive plants would experience higher competitive pressures [40]. Some studies also show that, due to the differentiation of species along different resource gradients, communities with high species diversity have higher structural complexity, which increases the resistance of native communities to plant invasions [35,41]. Similar to these findings, increasing species richness decreased *A*. *philoxeroides* coverage in this study. This is because habitats with abundant species usually have stronger interspecific interactions, and intensified competition would limit the establishment and spread of invasive plants [4,42,43]. Furthermore, high species richness can effectively resist plant invasion by increasing native plant biomass and increasing the difference in functional traits between invasive and native species [44,45].

Due to the heterogeneous responses of growth rate, allelopathic effect, and competitiveness between native and invasive plants to soil environments, soil properties also significantly affect the plant invasion progress [13,16,30]. Among them, soil potassium can promote protein synthesis and improve the water use efficiency of plants, and it can increase plant species diversity, which effectively resists plant invasions [46,47]. In our study, we also found a negative relationship between soil AK and *A*. *philoxeroides* coverage, possibly because AK may promote the growth of many accompanying plant species in invaded communities, and the competition of plants for potassium weakens the positive ‘invasive plant–soil’ feedback [48,49], thus leading to a decrease in *A*. *philoxeroides* invasion.

### 3.2. Mechanism of Impact of Soil on Plant Diversity

The Patrick richness index of the control plots had a maximum value at moderate CWC because CWC is closely related to the root distribution, moisture absorption, physiological metabolism, and electrolyte balance of plants; soil drought easily causes plant death due to water deficiency, while excessive soil moisture can weaken plant respiration and thus decrease plant richness [11,47]. In RDA, the effect of CWC on plant diversity was weak, perhaps because the interactions between multiple soil factors masked the determinant of CWC. Regression analysis and RDA all showed that the increasing N-NH_4_ significantly decreased the Patrick richness index, the Simpson dominance index, and the Shannon–Wiener diversity index because high N-NH_4_ caused toxicity in seed germination, seedling growth, and tillering and thus inhibited plant survival [43]. Excessive N-NH_4_ can also disrupt the balance of alkaline cations of plants, causing indirect toxicity by accelerating the replacement of soil Ca^+^ and Mg^+^ ions [50]. In addition, high N-NH_4_ enhances the nitrogen-scavenging activity of soil microorganisms and decreases the biomass and secretions of plant roots, which might decrease plant diversity [51,52]. The Simpson dominance index had a strong negative relationship with soil AP in the control plots because high AP decreases the activity of soil microorganisms and acid phosphatase, which decelerate nutrient cycling, weaken plant nutrient utilization efficiency, and cause a decrease in plant diversity [52,53]. A similar result was found in the plant communities of the Ebinur Lake watershed [54].

None of the soil factors in the invaded plots determined the RDA axis 1 due to plant invasions disturbing the inherent community structure and the feedback between native plants and soil, which weakens the ecological effect of the soil [55]. The water-holding and retention functions of soil are dependent on the distribution and connectivity of soil pores, among which, non-capillary pores provide channels for the infiltration and exchange of air, water, and nutrients close to plant roots, while capillary pores provide soil with water storage functions [15,47,56]. In the invaded plots in our study, the Pielou evenness index had a positive relationship with NCP and a negative relationship with CP. This is because the increasing NCP provided more underground niches for most plant roots, which alleviated the interspecific competition of *A*. *philoxeroides*-invaded communities and increased the Pielou evenness index [57]. Although increasing CP can decrease soil saturated hydraulic conductivity, invasive plants, as ‘opportunity species’, usually have strong phenotypic plasticity and ecological adaptability, which increases the invaders’ tolerance to changes in soil microhabitats compared with native accompanying plants and achieves greater competitiveness [16,48,58]. This intensifies the exclusion of *A*. *philoxeroides* against native plants and thus decreases species evenness. Moreover, compared with the control plots, the determinants of plant diversity in the invaded plots were mostly soil physical indicators, possibly because *A*. *philoxeroides*, as the constructive species, had strong adaptability to soil nutrient fluctuations, which weakened the impact of soil chemical indicators on the species diversity of the invaded communities [22,59].

Notably, our study area is located on the Qinling–Huaihe line, which is the geographical boundary between North and South China and also a climatic transition zone from a subtropical climate to a warm temperate climate. This zone has complicated variations in climate, terrain, and rainfall; these factors can significantly affect soil physicochemical properties [60], and all of them constitute a high degree of habitat heterogeneity that provides favorable conditions for the systematic development, niche differentiation, and species coexistence of plants in the study area, resulting in a high native plant diversity for resisting plant invasions [61,62]. However, the fluctuating temperature and precipitation in the climatic transition zone have caused lots of alien plants to respond more positively than native plants, which can enhance their ecological plasticity and interspecific interactions, thus accelerating the spread of many invasive plants, including *A*. *philoxeroides*, in the study area [63,64]. Therefore, under rapid global climate change, we should intensify plant diversity protection and invasion assessment in the North–South climatic transition zone of China.

### 3.3. Mechanism of Impact of Soil on Plant Species Distributions

Soil pore structure affects the distribution, adsorption, and release of nutrients within or between soil aggregates, while soil moisture conditions can regulate the absorption efficiency and leaching process of soil nutrients by plants [56,65]. Most plant species in the control plots were distributed in areas with low TP and FWC, which was related to their own properties and soil conditions. Consistent with our results, some studies have found that many native weeds prefer to distribute in low-porosity habitats, and some have extremely strong drought resistance, which leads to their shallower roots and lower water demand, making them accumulate in soil with lower water capacity [14,66,67]. Moreover, other related studies have found that a moderate increase in soil compaction can stimulate herbaceous plants to produce high-density fine roots in order to form channels, thus increasing soil ventilation and permeability and enabling plants to better adapt to low-FWC habitats [68,69]. *I*. *cylindrical* and *C*. *dactylon*, which belong to the Poaceae family, develop root systems that can increase the sand content of soil and indirectly enhance soil cohesion; however, their root growth consumes a large amount of water, so they tend to distribute in high-soil CP habitats with strong water storage capacities [14,70]. *D*. *indica* has a high nitrogen absorption rate, while N-NO_3_, as the main nitrogen source, can promote the lateral root growth of *D*. *indica* and participate in metabolic regulation [71,72]; thus, *D*. *indica* positively correlates with N-NO_3_. *O*. *articulata* has thick roots and mainly relies on bulb propagation, which prompts it to distribute in loose and porous soil environments with high ORG [36,37].

*A*. *philoxeroides* was located at the center of the CCA diagram, indicating that this invader has high adaptability to various soil factors, which is not only due to its strong phenotypic plasticity, reproductive ability, and stress resistance [24,38] but also its regulation by the ‘soil–plant’ feedback effect, which further promotes *A*. *philoxeroides* expansion [28,29]. *C*. *rotundus* positively correlates with N-NH_4_ due to the mycorrhizal fungi in its root systems; these mycorrhizal fungi can increase the absorption of soil N-NH_4_ and transfer it to the host plant [73,74]. *P*. *paspaloides* and *H*. *sibthorpioides* were weakly affected by soil factors due to their wet and shade-tolerant properties, as well as their vegetative and sexual reproduction, which decreased the limiting effects of soil environments on these two plant species [75]. For *P*. *paspaloides* especially, as the most dominant accompanying species of aquatic ecotype *A*. *philoxeroides*, its demand for water environments is higher than for soil environments [22]. *O*. *articulata* was distributed in a location very close to *A*. *philoxeroides*, indicating that the ecological niche of these two plant species is similar, and they thus might be intensely competitive. Our previous study also found that *O*. *articulata* has a strong biotic substitution effect on *A*. *philoxeroides* invasion, particularly in soils with high fertility; the physiological plasticity and photosynthetic efficiency of *O*. *articulata* were higher than those of *A*. *philoxeroides* [76].

However, plant invasions can also affect biotic community structures by altering soil physicochemical properties [59]. In this study, some plant species, such as *A*. *argyi*, *D*. *micrantha*, and *B*. *syzigachne*, tended to be distributed in habitats with low soil porosity and water capacity, possibly because *A*. *philoxeroides* invasion decreases the retention of soil nutrition and moisture and causes a decrease in soil permeability [77]. Furthermore, a small input from *A*. *philoxeroides* can increase soil nitrogen cycling in invaded habitats [12,48]. All of these phenomena enhanced the small-scale soil resource heterogeneity and finally exacerbated the patchy plant distribution in the invaded communities.

## 4. Materials and Methods

### 4.1. Field Survey

Our study was conducted in Xinyang City, Southern Henan Province, China. Xinyang City is located in the upper reaches of the Huai River on the Qinling Mountains–Huaihe River Line (Qinling–Huaihe Line), which is the geographical boundary between North and South China. Xinyang City is in a transitional zone from a subtropical climate to a warm temperate climate, and it has high plant species diversity within this region. Our study area has an average annual sunshine time of 2000 h, an average annual temperature of 15.2 °C, an average frost-free period of 225 d, and an average annual precipitation of 1105 mm, and the main soils are paddy soil and yellow-brown soil [78]. Due to global changes, the number of invasive plants in Xinyang City has significantly increased, among which, *A*. *philoxeroides* has widely invaded multiple ecosystems [64].

During the vigorous period of plant growth from June to July of 2022, we selected habitats with a continuous terrestrial *A*. *philoxeroides* invasion area of over 50 m^2^ to set invaded sampling plots in Xinyang City. In total, we set 30 *A*. *philoxeroides*-invaded plots, and every 2 plots was more than 5 km apart (Figure 6). We also set a total of 30 control plots (native plant communities without *A*. *philoxeroides* invasion) in areas with similar habitats near each invaded plot. Each plot had an area of 5 m × 5 m. We evenly set three 5 m transects in each plot and then evenly set five quadrats with an area of 0.5 m × 0.5 m along each transect for plant community investigation (Figure 6). We recorded the species name, abundance, individual height, and coverage of all plant species in each sampling plot. We recorded the number of asexual branches (for clonal plants), tillers (for Poaceae plants), or individuals (for non-clonal plants) as the abundance measurement for each plant species. We defined plant height as the vertical distance from the soil surface to the tip of the highest branch or longest leaf and measured the average height of 10 randomly selected individuals by using steel tape (if there were <10 individuals in a species, we measured all of its individuals). We used a 0.5 m × 0.5 m metal frame with 100 cells to calculate the coverage of plants [22].

After completing the plant community investigation, we evenly set three soil sampling sites along an arbitrary diagonal in each plot using ring knives, each with a volume of 100 cm^3^, to collect in situ soil from the 0–20 cm soil layer. We then brought these ring knives carrying in situ soil back to the laboratory to measure 6 soil physical indicators, namely, bulk density (BD), full water capacity (FWC), capillary water capacity (CWC), non-capillary porosity (NCP), capillary porosity (CP), and total porosity (TP). We also used the five-point sampling method to collect soil samples from a 0–20 cm soil layer, evenly mixed soil samples from the same plot into one part, and then put them into plastic bags separately. After drying and sieving, we measured 6 soil chemical indicators, namely, ammonium nitrogen (N-NH_4_), nitrate nitrogen (N-NO_3_), available phosphorus (AP), available potassium (AK), organic matter (ORG), and pH value.

### 4.2. Indicator Measurement

#### 4.2.1. Measurement of Soil Physical Properties

We put each ring knife carrying in situ soil (without the top and bottom covers) in a flat-bottomed container, added water to the container until it was flush with the upper edge of the ring knife, and let the ring knife soak in water for several hours until the soil was saturated with water. We then removed the ring knife from the container, wiped off any external water, and covered the top and bottom covers; the total weight of the ring knife and soils in this first state was recorded as W_1_. We then opened the top and bottom covers (retaining the mesh) and placed the ring knife on the bracket for 12 h to facilitate the drainage of gravity water from the in situ soil and then covered the top and bottom covers again; the total weight in this second state was recorded as W_2_. We then opened the top and bottom covers and placed the in situ soil in a ring knife that carried it into a drying oven (Leirun 101-4A, China) set to 105 °C, and it remained there until it reached a constant weight; the total weight in this third state was recorded as W_3_. We also recorded the weight of an empty ring knife as W_0_ and the volume of a ring knife as V. Based on the above data, we could further calculate the 6 soil physical indicators [79].

#### 4.2.2. Measurement of Soil Chemical Properties

The inorganic nitrogen of the soil samples was extracted at 2 mol/L KCl. We then measured soil N-NH_4_ and N-NO_3_ by using an auto discrete analyzer (Easychem Plus, Systea S.p.A., Rome, Italy). We measured soil AP and AK by using a continuous flow analyzer (Futura, AMS Alliance, Paris, France). We measured the total soil organic carbon by using a vario TOC cube analyzer (Elementar Analysensysteme GmbH, Hanau, Germany) and multiplied it by 1.724 to obtain the value of soil ORG. We mixed the soil samples with deionized water in a 1:5 ratio (weight–volume) and measured the soil pH value by using an intelligent acidity meter (Mettler-Toledo International Inc., Zurich, Switzerland) [22].

### 4.3. Data Calculation

#### 4.3.1. Soil Physical Indicators

Based on the data measured with the ring knife and water immersion methods, as described in Section 4.2.1, the 6 soil physical indicators were calculated by using the following formulas [80]:BD = (W_3_ − W_0_)/V;
FWC = (W_1_ − W_3_)/(W_3_ − W_0_) × 100%;
CWC = (W_2_ − W_3_)/(W_3_ − W_0_) × 100%;
NCP = (W_1_ − W_2_)/V × 100%;
CP = (W_2_ − W_3_)/V × 100%;
TP = (W_1_ − W_3_)/V × 100%.

#### 4.3.2. Importance Value

The importance value (IV) is a comprehensive indicator that measures the status and role of a species in communities; the larger the IV, the higher the dominance of this species. The relative IV was calculated by using the following formula [38]:Relative IV = (relative height + relative coverage + relative abundance)/3
where relative height, relative coverage, and relative abundance were the percentages of a species’ average height, coverage, and abundance over the sum of all species’ average height, coverage, and abundance within a plot, respectively.

The total IV of a species was the sum of its relative IV in all sampling plots.

#### 4.3.3. Species Diversity

The following 4 α-species diversity indices were calculated to assess the species diversity of *A*. *philoxeroides*-invaded and native plant communities [22]:Patrick richness index: *R* = S;
Simpson dominance index: *λ* = 1 − ΣP_i_^2^;
Shannon–Wiener diversity index: *H* = −ΣP_i_ × lnP_i_;
Pielou evenness index: *E* = *H*/lnS.
where S is the total number of plant species in a plot, and P_i_ is the relative IV of species i.

### 4.4. Statistical Analyses

We conducted a regression analysis to examine the pairwise coupling relationships between soil physicochemical indicators, species diversity index, and *A*. *philoxeroides* invasion coverage by using the SPSS 16.0 software (SPSS Inc., Chicago, IL, USA). This software provided 11 curve-fitting models, and we selected the significant regression model with the highest fitting coefficient for analysis. We established an environmental matrix containing 12 soil physicochemical indicators (30 × 12) and a species diversity matrix containing 4 diversity indices (30 × 4). We then conducted redundancy analysis (RDA) to examine the comprehensive impacts of soil factors on plant diversity by using the Canoco 4.5 software (Microcomputer Power, New York, NY, USA). We also established a relative IV matrix containing 25 dominant plant species in native communities (30 × 25) and 21 dominant plant species in invaded communities (30 × 21), and we conducted canonical correspondence analysis (CCA) to examine the plant distribution along the soil environmental gradient with the Canoco 4.5 software. A Monte Carlo permutation test based on 499 random permutations was performed to test the significance of the correlation coefficient between soil indicators and RDA axes and CCA axes. We drew the RDA and CCA ordination diagrams by using the CanoDraw function of the Canoco 4.5 software [22].

## 5. Conclusions

We found that *A*. *philoxeroides* invasion widened the gap between the importance values of dominant plant species. Some native plants with weak tolerance were pushed out of the dominant layer by *A*. *philoxeroides*, and the coverage of *A*. *philoxeroides* decreased with increasing soil AK and species richness. Soil chemical properties determined the species diversity of control plant communities but had a weak impact on that of *A*. *philoxeroides*-invaded plant communities, while the species evenness of the invaded plant communities was mainly affected by soil porosity. The determinants of species distribution in the control communities were soil porosity, N-NO_3_, ORG, and FWC, while the determinants in invaded communities were soil AK, BD, N-NH_4_, porosity, and water capacity. These results indicate that *A*. *philoxeroides* invasion intensifies the interspecific competition between plant species in invaded habitats, and the plant species diversity and distribution in control and invaded communities have different responses to soil physicochemical properties. Thus, we should intensify the monitoring of changes in soil properties in *A*. *philoxeroides*-invaded habitats and implement biotic replacement strategies based on the heterogeneous responses of plants to soil resource fluctuations. Our findings could provide important implications for elucidating the species coexistence mechanisms of invaded ecosystems and predicting plant invasions under global environmental changes.

## Figures and Tables

**Figure 1 plants-13-01196-f001:**
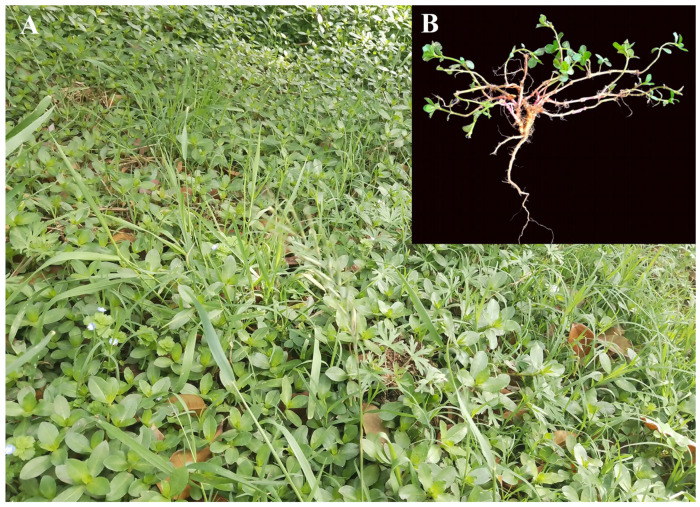
Terrestrial *A*. *philoxeroides*-invaded plant communities (**A**) and the studied species, *A*. *philoxeroides* (**B**), in Xinyang City, China (picture: Hao Wu).

**Figure 2 plants-13-01196-f002:**
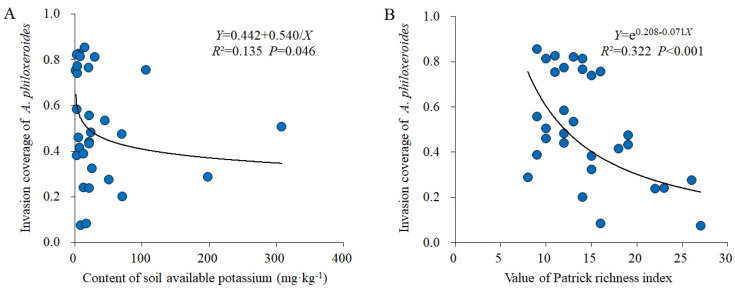
Regression analysis between *A*. *philoxeroides* invasion coverage and soil factors (**A**) and species diversity index (**B**).

**Figure 3 plants-13-01196-f003:**
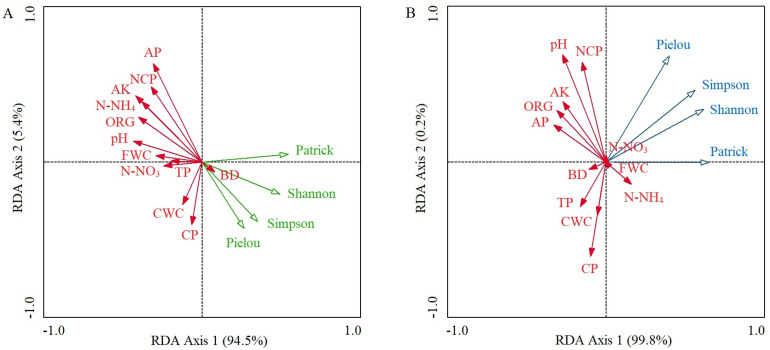
RDA ordination diagram of soil factors and species diversity index in control (**A**) and *A*. *philoxeroides*-invaded plant communities (**B**). Patrick, Simpson, Shannon, and Pielou represent the Patrick richness index, Simpson dominance index, Shannon–Wiener diversity index, and Pielou evenness index. Solid red vectors represent the 12 soil factors; hollow vectors represent 4 α-species diversity indices. Arrows indicate the direction of increase in variables from the RDA ordination center. The angle between each pair of variables represents their correlations; the smaller the angle, the greater the correlation.

**Figure 4 plants-13-01196-f004:**
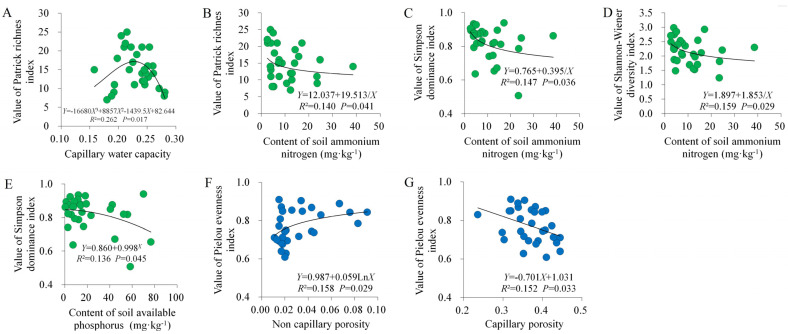
Regression analysis between soil factors and species diversity index in control (**A**–**E**) and *A*. *philoxeroides*-invaded plant communities (**F**,**G**).

**Figure 5 plants-13-01196-f005:**
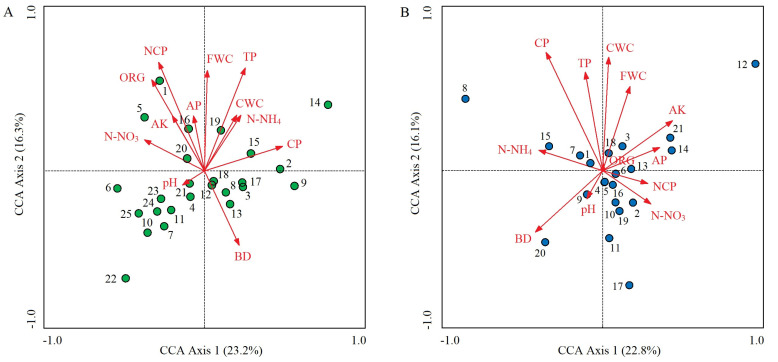
CCA ordination diagrams of dominant plant species in control (**A**) and *A*. *philoxeroides*-invaded plant communities (**B**) along soil property gradient. Red vectors represent the 12 soil factors. Solid dots represent the dominant plant species, and the locations represent their optimal distribution positions under multiple soil factor combinations. Arabic numbers are codes of dominant plant species, and their Latin names are shown in Table 1. Vertical distance between dots and vectors represents the influence degree of a certain soil factor on a certain species distribution; the shorter the vertical distance, the greater the influence degree.

**Figure 6 plants-13-01196-f006:**
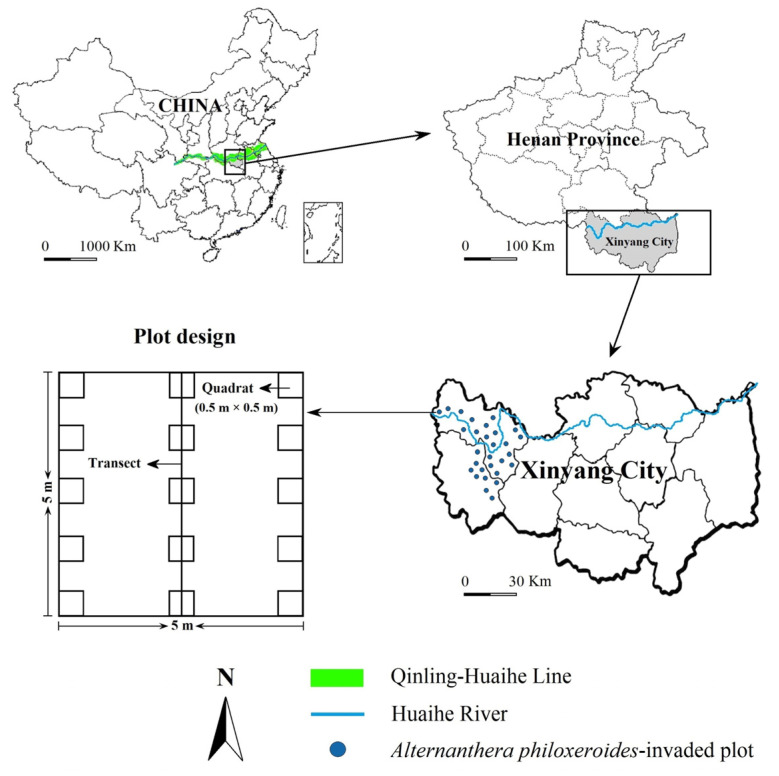
Sampling plots of terrestrial *A*. *philoxeroides*-invaded plant communities along the geographical boundary between North and South China. Plot size: 5 × 5 m; transect size: 5 m; quadrat size: 0.5 × 0.5 m.

**Table 1 plants-13-01196-t001:** Total importance value (IV) of dominant species in control and *A*. *philoxeroides*-invaded plant communities.

Code	Plant Species	Total IV	Code	Plant Species	Total IV
Control communities			Invaded communities		
1	*Oxalis articulata*	4.030	1	*Alternanthera philoxeroides*	15.309
2	*Cynodon dactylon*	3.817	2	*Cynodon dactylon*	2.266
3	*Eleusine indica*	3.533	3	*Galium odoratum*	1.917
4	*Galium odoratum*	2.862	4	*Oxalis articulata*	1.915
5	*Duchesnea indica*	1.645	5	*Eleusine indica*	1.758
6	*Dichondra micrantha*	1.614	6	*Rumex acetosa*	1.243
7	*Axonopus compressus*	1.376	7	*Medicago sativa*	1.116
8	*Lolium perenne*	1.318	8	*Paspalum paspaloides*	1.078
9	*Digitaria sanguinalis*	1.234	9	*Erigeron annuus*	0.993
10	*Medicago sativa*	1.119	10	*Duchesnea indica*	0.978
11	*Poa annua*	1.057	11	*Artemisia argyi*	0.956
12	*Erigeron annuus*	1.029	12	*Hydrocotyle sibthorpioides*	0.824
13	*Vicia hirsuta*	0.823	13	*Poa annua*	0.822
14	*Kyllinga brevifolia*	0.765	14	*Ophiopogon bodinieri*	0.822
15	*Imperata cylindrica*	0.760	15	*Cyperus rotundus*	0.710
16	*Ophiopogon bodinieri*	0.732	16	*Lolium perenne*	0.655
17	*Cayratia japonica*	0.721	17	*Dichondra micrantha*	0.646
18	*Mazus japonicus*	0.711	18	*Cayratia japonica*	0.622
19	*Stellaria media*	0.700	19	*Imperata cylindrica*	0.543
20	*Veronica polita*	0.648	20	*Beckmannia syzigachne*	0.532
21	*Sedum sarmentosum*	0.616	21	*Mazus japonicus*	0.512
22	*Kummerowia striata*	0.529			
23	*Rumex acetosa*	0.526			
24	*Plantago asiatica*	0.518			
25	*Clinopodium chinense*	0.512			

**Table 2 plants-13-01196-t002:** Correlations between the soil physicochemical factors and the first two RDA axes.

Soil Factors	Control Communities		Invaded Communities	
	Axis 1	Axis 2	Axis 1	Axis 2
BD	0.081	−0.064	−0.107	−0.047
FWC	−0.290	0.041	0.009	0.011
CWC	−0.121	−0.273	−0.054	−0.333
NCP	−0.321	0.404 *	−0.150	0.629 **
CP	−0.064	−0.402 *	−0.097	−0.594 **
TP	−0.196	0.007	−0.162	−0.280
N-NH_4_	−0.379 *	0.388 *	0.162	−0.140
N-NO_3_	−0.241	−0.025	0.001	0.047
AP	−0.307	0.634 **	−0.329	0.235
AK	−0.418 *	0.426 *	−0.271	0.380 *
ORG	−0.400 *	0.289	−0.309	0.325
pH	−0.433 *	0.135	−0.273	0.677 **

BD, FWC, CWC, NCP, CP, TP, N-NH_4_, N-NO_3_, AP, AK, ORG, and pH represent bulk density, full water capacity, capillary water capacity, non-capillary porosity, capillary porosity, total porosity, ammonium nitrogen, nitrate nitrogen, available phosphorus, available potassium, organic matter, and pH value, respectively. * *p* < 0.05 level; ** *p* < 0.01 level. The same applies below.

**Table 3 plants-13-01196-t003:** Correlations between the soil physicochemical factors and the first two CCA axes.

Soil Factors	Control Communities		Invaded Communities	
	Axis 1	Axis 2	Axis 1	Axis 2
BD	0.267	−0.573 **	−0.419 *	−0.383 *
FWC	0.022	0.772 **	0.171	0.525 **
CWC	0.248	0.429 *	0.038	0.706 **
NCP	−0.352	0.835 **	0.280	−0.080
CP	0.601 **	0.189	−0.352	0.737 **
TP	0.314	0.791 **	−0.109	0.614 **
N-NH_4_	0.281	0.428 *	−0.398 *	0.127
N-NO_3_	−0.463 **	0.238	0.300	−0.207
AP	−0.084	0.422 *	0.355	0.145
AK	−0.247	0.423 *	0.435 *	0.311
ORG	−0.406 *	0.701 **	0.004	0.046
pH	−0.168	−0.108	−0.098	−0.172

* *p* < 0.05 level; ** *p* < 0.01 level.

## Data Availability

Data will be made available upon request.

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
