# Peer review of "Impacts of Soil Properties on Species Diversity and Structure in Alternanthera philoxeroides-Invaded and Native Plant Communities"

_plants, 2024, doi:10.3390/plants13091196_

Round 1

Reviewer 1 Report

Comments and Suggestions for Authors

The authors compared the soil properties and the vegetation in the Alternanthera philoxeroides-invaded plots and the native plant plots, then proposed the change of soil properties for the appearance of A. philoxeroides. The results found the species diversity and distribution have different response to soil properties. The descriptions and the results are fine and well presented. There are only some small questions:

1. The figures quality is quite worse. The resolution of all figures need to be improved. Especially for fig. 3, all the numbers and the symbols are hard to see. The 0.5 m in fig. 5 is also difficult to distinguished. For the figure should present after the context, figure 5 should move to the M&M sections.

2. The authors emphasis the sampling plots are located on the geographical boundary between N. and S. China, especially between subtropical to warm temperate climate, however, the authors have no any discussion about the impact of such geographic distribution to the species diversity or the soil properties, also the impact for the invasion of A. philoxeroides. For the geographical gradients, the influence of temperature and precipitation on the soil properties should be also considered.

3. The 15 sampling quadrat were selected in a sampling plot. Why the authors divided the plots to so small and many quadrats? Some herbal species could grow very large and one tussock could occupy large area. The Line 385-387 not really descript the plant survey were only in the quadrats and merge the data or sampling all the plot. Also how to decide the height of each species? and how the relative height calculated? How to decide the abundance? Is it calculated by individuals? It’s normally very difficult to decide when the grass grow as a tussock or grow by asexual reproduction. Also it influences the results of IV greatly. The authors should make more clear descriptions.

4. The calculation of diversity is dependent by IV in this work. For many study case for the grass, meadow and marshland, mostly the diversity index were only calculated by the relative coverage of each species. The authors should descript and consider the influence of the different methods.

Author Response

Dear editor,

We sincerely appreciate the editor and reviewers’ comments, which are very useful for polishing the earlier version of this manuscript. We carefully reexamined our manuscript and made revisions according to these comments. We addressed the comments by reviewers and lay out how we had addressed these issues, point by point, and all changes are displayed by using the red fonts in the revised version.

For reviewer 1:

  1. The figures quality is quite worse. The resolution of all figures need to be improved. Especially for fig. 3, all the numbers and the symbols are hard to see. The 0.5 m in fig. 5 is also difficult to distinguished. For the figure should present after the context, figure 5 should move to the M&M sections.

Response: For improving the figures quality, we have inserted the original drawings in the revised version, and provided original drawing materials in attachment for reediting. For fig. 3 (fig. 4 in revision), we have improved the size of numbers and the symbols. For fig. 5 (fig. 6 in revision), we have improved the font size of 0.5 m × 0.5 m. We have move the fig. 5 (fig. 6 in revision) to the M&M sections as suggested.

  1. The authors emphasis the sampling plots are located on the geographical boundary between N. and S. China, especially between subtropical to warm temperate climate, however, the authors have no any discussion about the impact of such geographic distribution to the species diversity or the soil properties, also the impact for the invasion of A. philoxeroides. For the geographical gradients, the influence of temperature and precipitation on the soil properties should be also considered.

Response: Notably, our study area is located in the Qinling–Huaihe line, which is the geo-graphical boundary between North and South China, and also a climatic transition zone from subtropical to warm temperate climate. This zone has complicated variations of climate, terrain, and rainfall, while these factors can significantly affect the soil physicochemical properties(Kou et al., 2020), and all of these constitute a high degree of habitat heterogeneity which providing favorable conditions for the systematic development, niche differentiation, and species coexistence of plants in the study area, resulting in a high native plant diversity for resisting plant invasions (Zhang et al., 2021; Zhou et al., 2021). However, the fluctuating temperature and precipitation in the climatic transition zone have caused lots of alien plants responding more positively than native plants, which would enhance their ecological plasticity and interspecific interaction intensity, and thus accelerates the spread of many invasive plants including A. philoxeroides in the study area (Wallingford et al., 2020; Yan et al., 2020). Therefore, under a rapid global climate change, we should intensify plant diversity protection and invasion assessment in the climatic transition zone of China. We have added the above contents to the revision as the reviewer suggested.

  1. The 15 sampling quadrat were selected in a sampling plot. Why the authors divided the plots to so small and many quadrats? Some herbal species could grow very large and one tussock could occupy large area. The Line 385-387 not really descript the plant survey were only in the quadrats and merge the data or sampling all the plot. Also how to decide the height of each species? and how the relative height calculated? How to decide the abundance? Is it calculated by individuals? It’s normally very difficult to decide when the grass grow as a tussock or grow by asexual reproduction. Also it influences the results of IV greatly. The authors should make more clear descriptions.

Response: We use a lot of small quadrats to record as many plant species as possible in the sampling plot, for more accurately evaluating species diversity of the investigated plant community. Just as the reviewer stated that some herbal species (such as Alternanthera philoxeroides) may occupy large areas, however, our 15 quadrats are evenly set in each plot, so they could objectively reflected the dominance of certain plant species (the greater the coverage occupied in each quadrat, the higher the dominance).

We defined plant height as the vertical distance from soil surface to the tip of the highest branch or longest leaf, and measured the average height of 10 randomly selected individuals by using a steel tape (if individuals of a certain species were <10, we measured all individuals). Relative height referred to the percentages of one species’ average height over the sum of all species average height within a plot.

About the abundance, we conducted different counting methods for different types of plants. We recorded numbers of asexual branches (for clonal plants), tillers (for Poaceae plants), or individuals (for non-clonal plants) as the abundance measure for each plant species. And we used a 0.5 m × 0.5 m metal frame with 100 cells to calculate the coverage of plants.

    We have added above information to revised manuscript as reviewer suggested.

  1. The calculation of diversity is dependent by IV in this work. For many study case for the grass, meadow and marshland, mostly the diversity index were only calculated by the relative coverage of each species. The authors should descript and consider the influence of the different methods.

Response: Many studies abut herbaceous community indeed calculated the diversity index only dependent on relative coverage as the reviewer stated, however, it may not be applicable for our research. Because our aim is to analyze the differences between Alternanthera philoxeroides-invaded community and the control community, in invaded community, some native plant species could coexist with the invader A. philoxeroides actually benefiting from their large plant height (such as Artemisia argyi which belonging to Asteraceae) and abundance (such as Cynodon dactylon, Eleusine indica and Paspalum paspaloides, which belonging to Poaceae and having the tillering growth). If we only used the coverage for calculating diversity index without considering height and abundance, then, this would artificially weaken the dominance and effect of native accompanying plant species, and thus disrupted accurately evaluating the plant species diversity of A. philoxeroides-invaded community. In summary, using the importance value (IV, a comprehensive indicator that covering height, coverage, and abundance) to calculate 4 alpha diversity indices in our study is perhaps more suitable.

Reviewer 2 Report

Comments and Suggestions for Authors

Reviewer(s)' Comments to Author:
Manuscript ID: plants-2945830

Title: Impacts of Soil Properties on Species Diversity and Distribution in Alternanthera philoxeroides-Invaded and Native Plant Communities

Authors: Hao Wu, Yu-xin Liu, Tian-tian Zhang, Ming-xia Xu and Ben-qiang Rao

First of all, congratulations, very nice article, complex work. Nowadays, in the age of meta-analyses performed on large databases, the work from own data collection is certainly noteworthy.

The different impacts of soil physicochemical properties on species diversity are extremely diverse and complex process.

At the same time, precisely because of its complexity, it is sometimes difficult to interpret, or at least clearer definitions are needed. The "Distribution" mentioned in the title does not even appear with enough emphasis in the article, so I recommend changing the title or giving a more detailed explanation.

The article’s main strength is that it collects, organizes, and compares the elements of the difficult-to-comparable, soil properties and different diversity methods use along various parameters with systematic sampling.

This is an important message, that soil chemical properties determined the species diversity of control plant communities, but had a weak impact on that of A. philoxeroides-invaded plant communities.

The aims, sampling procedure, methods of data analysis are clearly stated and introduced. The statistical analyses are appropriate. The results and facts are presented clearly and sufficiently fully and are separated from interpretations. The authors know well the literature of the subject, and fairly discuss the correspondence of results.

Nevertheless, the message of the research should be better focused. This could be reinforced with a summary table, where you really only see the end result, what affects what, and how.

Further remarks and minor mistakes in the order of the text:

The abstract is good, concise and understandable.

I recommend entering new keywords, because the current ones can all be found in the title, for example „soil properties”.  This way the new, changed keyword will help others find the article.

Introduction:

in line 58.: in name Gaillardia aristata instead of G. alistata

in line 82.: I suggest that Alternanthera philoxeroides, one of the main species of the paper should be a more detailed presentation. What Family does it belong to, and what are its most important morphological and physiological characteristics that influence the successful spread of the species? I would appreciate a schematic draw or photograph about the studied species and the investigatid vegetation.

Results

in line 220.: Table 3. It is recommended to switch the order of „Control communities” and „Invaded communities”.

Materials and Methods:

in line 376.: „belong to the Asteraceae, Leguminosae, Poaceae, and Amaranthaceae families” This enumeration is unnecessary here.

Author Response

Dear editor,

We sincerely appreciate the editor and reviewers’ comments, which are very useful for polishing the earlier version of this manuscript. We carefully reexamined our manuscript and made revisions according to these comments. We addressed the comments by reviewers and lay out how we had addressed these issues, point by point, and all changes are displayed by using the red fonts in the revised version.

For reviewer 2:

The "Distribution" mentioned in the title does not even appear with enough emphasis in the article, so I recommend changing the title or giving a more detailed explanation.

Response: We have changed the ‘Distribution’ to ‘Structure’ in the title as the reviewer suggested, as the ‘Structure’ could simultaneously include the species composition and distribution that mentioned in our article.

Nevertheless, the message of the research should be better focused. This could be reinforced with a summary table, where you really only see the end result, what affects what, and how.

Response: Thanks for reviewer’s constructive suggestions, a summary table may clearly show the end result. However, in our study, we examine the impact of soil properties on species diversity in two different types of plant communities based on both univariate analysis (regression) and multivariate analysis (RDA). Interestingly, we have found that the results obtained from these two statistic methods are not completely consistent, such as the impact of CWC on diversity in control community and the impact of pH on diversity in invaded community. This also indicates that compared to single factor effects, there are also complex interactions among multiple soil factors, thus the framework table may not be able to accurately show what soil factor affects what diversity index and how.

I recommend entering new keywords, because the current ones can all be found in the title, for example, soil properties”.  This way the new, changed keyword will help others find the article.

Response: According to the reviewer’s suggestions, we change ‘soil properties’ to ‘habitat heterogeneity’, change ‘plant invasions’ to ‘biological invasions’, change ‘community structure’ to ‘plant community’, and change ‘species diversity’ to ‘diversity index’, for refusing repetitions with the title. We also added a new keyword of ‘species coexistence’, for helping others find our article.

in line 58.: in name Gaillardia aristata instead of G. alistata

Response: We changed ‘Gaillardia alistata’ to ‘Gaillardia aristata’ as suggested.

in line 82.: I suggest that Alternanthera philoxeroides, one of the main species of the paper should be a more detailed presentation. What Family does it belong to, and what are its most important morphological and physiological characteristics that influence the successful spread of the species? I would appreciate a schematic draw or photograph about the studied species and the investigatid vegetation.

Response: Alternanthera philoxeroides belongs to the Amaranthaceae family, and has the robust asexual reproductive ability, which enables it to propagate asexually through the fracture of stem segments and adventitious roots on stem nodes, and thus rapidly spreads and establishes populations across diverse habitats. Furthermore, A. philoxeroides can produce a variety of secondary metabolites, including alkaloids, organic acids and flavonoids, to inhibit the growth of other accompanying plants, providing favorable conditions for its diffusion and establishment. We have added above contents to the revised manuscript as the reviewer suggested and correspondingly modified the cited references [21].

We also have added the photograph of A. philoxeroides-invaded community and the studied species A. philoxeroides (Figure 1 of the revised version) as suggested, and have renumbered the figures throughout the text.

Results:

in line 220.: Table 3. It is recommended to switch the order of „Control communities” and „Invaded communities”.

Response: Due to the tables and figures in our manuscript are all arranged in the order of control- and invaded-communities, for clearly displaying the changes of community structures that caused by the A. philoxeroides invasion, thus it would be better to keep the original order in Table 3.

Materials and Methods:

in line 376.: „belong to the Asteraceae, Leguminosae, Poaceae, and Amaranthaceae families” This enumeration is unnecessary here.

Response: We have deleted these redundant contents as suggested.
